# Mining GOLD Samples for Conditional GANs

**Sangwoo Mo**[*]
KAIST
swmo@kaist.ac.kr

**Chiheon Kim**
Kakao Brain
chiheon.kim@kakaobrain.com

**Sungwoong Kim**
Kakao Brain
swkim@kakaobrain.com

**Minsu Cho**
POSTECH
mscho@postech.ac.kr

**Jinwoo Shin**
KAIST, AItrics
jinwoos@kaist.ac.kr

## Abstract

Conditional generative adversarial networks (cGANs) have gained a considerable attention in recent years due to its class-wise controllability and superior quality for complex generation tasks. We introduce a simple yet effective approach to improving cGANs by measuring the discrepancy between the data distribution and the model distribution on given samples. The proposed measure, coined the *gap of log-densities* (GOLD), provides an effective self-diagnosis for cGANs while being efficiently computed from the discriminator. We propose three applications of the GOLD: example re-weighting, rejection sampling, and active learning, which improve the training, inference, and data selection of cGANs, respectively. Our experimental results demonstrate that the proposed methods outperform corresponding baselines for all three applications on different image datasets.

## 1   Introduction

The generative adversarial network (GAN) [15] is arguably the most successful generative model in recent years, which have shown a remarkable progress across a broad range of applications, *e.g.*, image synthesis [5, 21, 40], data augmentation [49, 18] and style transfer [58, 10, 34]. In particular, as its advanced variant, the conditional GANs (cGANs) [31] have gained a considerable attention due to its class-wise controllability [9, 42, 10] and superior quality for complex generation tasks [39, 33, 5]. Training GANs (including cGANs), however, are known to be often hard and highly unstable [46]. Numerous techniques have thus been proposed to tackle the issue from different angles, *e.g.*, improving architectures [32, 56, 7], losses and regularizers  [16, 38, 20] and other training heuristics [46, 51, 8]. One promising direction for improving GANs would be to make GANs diagnose their own training and prescribe proper remedies. This is related to another branch of research on evaluating the performance of GANs, *i.e.*, measuring the discrepancy of the data distribution and the model distribution. One may utilize the measure to quantify better models [29] or directly use it as an objective function to optimize [37, 1]. However, measuring the discrepancy of GANs (and cGANs) is another challenging problem, since the data distribution remains unknown and the distribution GANs learn is implicit [35]. Common approaches to the discrepancy measurement of GANs include estimating the variational bounds of statistical distances [37, 1] and using an external pre-trained network as a surrogate evaluator [46, 17, 45]. Most previous methods on this line focus on classic unconditional GANs (*i.e.*, data-only densities), whereas discrepancy measures specialized for cGANs (*i.e.*, data-attribute joint densities) have rarely been explored.

**Contribution.**   In this paper, we propose a novel discrepancy measure for cGANs, that estimates the *gap of log-densities* (GOLD) of data and model distributions on given samples, thus being called

---

[*]This work was done as an intern at Kakao Brain.

the GOLD estimator. We show that it decomposes into two terms, marginal and conditional ones, that can be efficiently computed by two branches of the discriminator of cGAN. The two terms represent generation quality and class accuracy of generated samples, respectively, and the overall estimator measures the quality of *conditional generation*. We also propose a simple heuristic to balance the two terms, considering suboptimality levels of the two branches.

We present three applications of the GOLD estimator: example re-weighting, rejection sampling, and active learning, which improve the training, inference, and data selection of cGANs, respectively. All proposed methods require only a few lines of modification of the original code. We conduct our experiments on various datasets including MNIST [25], SVHN [36] and CIFAR-10 [23], and show that the GOLD-based schemes improve over the corresponding baselines for all three applications. For example, the GOLD-based re-weighting and rejection sampling schemes improve the fitting capacity [41] of cGAN trained under SVHN from 74.43 to 76.71 (+3.06%) and 73.58 to 75.06 (+2.01%), respectively. The GOLD-based active learning strategy improves the fitting capacity of cGAN trained under MNIST from 92.65 to 94.60 (+2.10%).

**Organization.** In Section 2, we briefly revisit cGAN models. In Section 3, we propose our main method, the gap of log-densities (GOLD) and its applications. In Section 4, we present the experimental results. Finally, in Section 5, we discuss more related work and conclude this paper.

## 2   Preliminary: Conditional GANs

The goal of cGANs is to learn the model distribution $p_g(x, c)$ to match with the attribute-augmented data distribution $p_{\text{data}}(x, c)$. To this end, a variety of architectures have been proposed to incorporate additional attributes [31, 46, 39, 57, 33]. The generator $G : (z, c) \mapsto x$ maps a pair of a latent $z$ and an attribute $c$ to generate a sample $x$ whereas the discriminator $D$ guides the generator to learn the joint distribution $p(x, c)$. Typically, there are two ways to use the attribute information: (a) providing it as an additional input to the discriminator (*i.e.*, $D : (x, c) \mapsto \{\text{real/generated}\}$) [31, 33], or (b) using it to train an auxiliary classifier for the attribute (*i.e.*, $D : x \mapsto (\{\text{real/generated}\}, c)$) [46, 39, 57]. The main difference between the two approaches can be viewed as whether to directly learn the joint distribution $p(x, c)$ or to separately learn the marginal $p(x)$ and the conditional $p(c|x)$.[2]

In this paper, we address training cGANs in a semi-supervised setting where a large amount of unlabeled data are available with only a small amount of labeled data. It is more attractive and practical than a fully-supervised setting in the sense that labeling attributes of all samples is often expensive while unlabeled data can be easily obtained. It is thus natural to utilize unlabeled data for improving the model, *e.g.*, via semi-supervised learning and active learning (see Section 3.2). While both of the two approaches above, (a) and (b), can be used in a semi-supervised setting, cGANs of (b) provide a more natural framework for using both labeled and unlabeled data;[3] one can use the unlabeled data to learn $p(x)$, and the labeled data to learn both $p(x)$ and $p(c|x)$. Therefore, we focus on evaluating the second type of architectures, *e.g.*, the auxiliary classifier GAN (ACGAN) [39]. We remark that our main idea in this paper is applicable to both types of cGANs in general.

The ACGAN model consists of the generator $G : (z, c) \mapsto x$ and the discriminator $D : x \mapsto (\{\text{real/generated}\}, c)$ consisting of the real/generated part $D_G : x \mapsto \{\text{real/generated}\}$ and the auxiliary classifier part $D_C : x \mapsto c$. Then, ACGAN is trained by optimizing both the GAN loss $\mathcal{L}_{\text{GAN}}$ and the auxiliary classifier loss $\mathcal{L}_{\text{AC}}$:

$$\mathcal{L}_{\text{GAN}} = \mathbb{E}_{(x,c) \sim p_{\text{data}}(x,c)}[-\log D_G(x)] + \mathbb{E}_{(z,c) \sim p_g(z,c)}[\log D_G(G(z,c))],$$
$$\mathcal{L}_{\text{AC}} = \mathbb{E}_{(x,c) \sim p_{\text{data}}(x,c)}[-\log D_C(c|x)] + \lambda_c \mathbb{E}_{(z,c) \sim p_g(z,c)}[-\log D_C(c|G(z,c))],$$

$$(1)$$

where $\lambda_c \geq 0$ is a hyper-parameter. Here, the generator and the discriminator minimize $-\mathcal{L}_{\text{GAN}} + \mathcal{L}_{\text{AC}}$ and $\mathcal{L}_{\text{GAN}} + \mathcal{L}_{\text{AC}}$, respectively.[4] The original work [39] simply sets $\lambda_c = 1$, but we empirically observe that using a smaller value often improves the performance: it strengthens the wrong signal of the generator when the generator produces bad samples with incorrect attributes. Such an issue has also been reported in related work of AMGAN[57] where the authors thus use $\lambda_c = 0$. On the other hand,

under a small amount of labeled data, a strictly positive value $\lambda_c > 0$ can be effective as it provides an effect of data augmentation to train the classifier $D_C$. In our experiments, we indeed observe that using a proper value (*e.g.*, $\lambda_c = 0.1$) improves the performance of ACGAN depending on datasets.

# 3 Gap of Log-Densities (GOLD)

In this section, we introduce a general formula of the gap of log-densities (GOLD) that measures the discrepancy between the data distribution and the model distribution on given samples. We then propose three applications: example re-weighting, rejection sampling, and active learning.

## 3.1 GOLD estimator: Measuring the discrepancy of cGANs

While cGANs can converge to the true joint distribution theoretically [15, 37], they are often far from being optimal in practice, particularly when trained with limited labels. The degree of suboptimality can be measured by the discrepancy between the true distribution $p_{\text{data}}(x, c)$ and the model distribution $p_g(x, c)$. Here, we consider the *gap of log-densities* (GOLD)[5], $\log p_{\text{data}}(x, c) - \log p_g(x, c)$, which can be rewritten as the sum of two log-ratio terms, marginal and conditional ones:

$$\log p_{\text{data}}(x, c) - \log p_g(x, c) = \underbrace{\log \frac{p_{\text{data}}(x)}{p_g(x)}}_{\text{marginal}} + \underbrace{\log \frac{p_{\text{data}}(c|x)}{p_g(c|x)}}_{\text{conditional}}. \tag{2}$$

Recall that cGANs are designed to achieve two goals jointly: generating a sample drawn from $p(x)$ and the distribution of its class is $p(c|x)$. The marginal and conditional terms measure the discrepancy on those two effects, respectively.

The exact computation of (2) is infeasible because we have no direct access to the true distribution and the implicit model distribution. Hence, we propose the GOLD estimator as follows. First, the marginal term $\log \frac{p_{\text{data}}(x)}{p_g(x)}$ is approximated by $\log \frac{D_G(x)}{1-D_G(x)}$ since the optimal discriminator $D_G^*$ satisfies $D_G^*(x) = \frac{p_{\text{data}}(x)}{p_{\text{data}}(x)+p_g(x)}$ [15]. Second, we estimate the conditional term $\log \frac{p_{\text{data}}(c|x)}{p_g(c|x)}$ using the classifier $D_C$ as follows. When a generated sample $x$ is given with its ground-truth label $c_x$, $p_g(c_x|x)$ is assumed be 1 and $p_{\text{data}}(c_x|x)$ is approximated by $D_C(c_x|x)$. When a real sample $x$ is given with the ground-truth label $c_x$, $p_{\text{data}}(c_x|x)$ is assumed to be 1 and $p_g(c_x|x)$ is approximated by $D_C(c_x|x)$. To sum up, the GOLD estimator can be defined as

$$d(x, c_x) := \begin{cases} \log \frac{D_G(x)}{1-D_G(x)} + \log D_C(c_x|x) & \text{if } x \text{ is a generated sample of class } c_x \\ \log \frac{D_G(x)}{1-D_G(x)} - \log D_C(c_x|x) & \text{if } x \text{ is a real sample of class } c_x \end{cases}. \tag{3}$$

Note that the conditional terms above for generated and real samples have opposite signs each other. This matches the signs of marginal and conditional terms for both generated and real samples as their marginal terms $\log \frac{D_G(x)}{1-D_G(x)}$ tend to be negative and positive, respectively.[6] Hence, (3) is reasonable to measure the joint quality of two effects of conditional generation.

For the derivation of (3), we assume the ideal (or optimal) discriminator $D^* = (D_G^*, D_C^*)$, which does not hold in practice. We often observe that the scale of marginal term is significantly larger than the conditional term because the density $p(x)$ is harder to learn than the class-predictive distribution $p(c|x)$ (see Figure 1a). This leads the GOLD estimator to be biased toward the generation part (marginal term), ignoring the class-condition part (conditional term). To address the imbalance issue, we develop a balanced variant of the GOLD estimator:

$$d_{\text{bal}}(x, c_x) := \begin{cases} \log \frac{D_G(x)}{1-D_G(x)} + \frac{\sigma_G}{\sigma_C} \log D_C(c_x|x) & \text{if } x \text{ is a generated sample of class } c_x \\ \log \frac{D_G(x)}{1-D_G(x)} - \frac{\sigma_G}{\sigma_C} \log D_C(c_x|x) & \text{if } x \text{ is a real sample of class } c_x \end{cases}, \tag{4}$$

where $\sigma_G$ and $\sigma_C$ are the standard deviations of marginal and conditional terms (among samples), respectively.

## 3.2 Applications of the GOLD estimator

**Example re-weighting.** A high value of the GOLD estimator suggests that the sample $(x, c_x)$ is under-estimated with respect to the joint distribution $p(x, c)$, and vice versa. Motivated by this, we propose an example re-weighting scheme for cGAN training, that guides the generator to focus on under-estimated samples during training. Formally, we consider the following re-weighted loss;

$$\mathcal{L}'_{\text{GAN}} = \mathbb{E}_{(x,c)\sim p_{\text{data}}(x,c)}[-\log D_G(x)] + \mathbb{E}_{(z,c)\sim p_g(z,c)}[d(G(z,c),c)^\beta \cdot \log D_G(G(z,c))],$$

$$\mathcal{L}'_{\text{AC}} = \mathbb{E}_{(x,c)\sim p_{\text{data}}(x,c)}[-\log D_C(c|x)] + \lambda_c \mathbb{E}_{(z,c)\sim p_g(z,c)}[-d(G(z,c),c)^\beta \cdot \log D_C(c|G(z,c))],$$

(5)

where $\beta \geq 0$ is a hyper-parameter to control the level of re-weighting and we use $x^\beta = -|x|^\beta$ for $x < 0$. Our intuition is that minimizing $\mathcal{L}'_{\text{GAN}} + \mathcal{L}'_{\text{AC}}$ encourages the discriminator $D$ to learn stronger feedbacks from the under-estimated (generated) samples, thus indirectly guiding the generator $G$ to emphasize their region. When the GOLD estimator $d(x, c_x)$ is negative, $D$ is trained to suppress the over-estimated samples, which indirectly regularizes $G$ to less focus on the corresponding region.

Since the GOLD estimator only becomes meaningful with sufficiently trained discriminators, we apply the re-weighting scheme with the loss of (5) after sufficiently training the model with the original loss of (1). We find that the GOLD estimator of generated samples stably converges to zero with the re-weighting scheme, while those only with the original loss do not converge (see Figure 1b). Note that one may also use the balanced version of the GOLD estimator $d_{\text{bal}}$ in (5). In our experiments, however, we simply use $d$ because $d_{\text{bal}}$ requires computing the standard deviations $\sigma_G$ and $\sigma_C$ along training, which significantly increases the computational burden. Improving the scheduling and/or re-weighting for training would be an interesting future direction.

**Rejection sampling.** Rejection sampling [44] is a useful technique to improve the inference of generative models, *i.e.*, the quality of generated samples. Instead of directly sampling from $p(x, c)$, we first obtain a sample from a (reasonably good) proposal distribution $q(x, c)$, and then accept it with probability $\frac{p(x,c)}{Mq(x,c)}$ for some constant $M > 0$ while rejecting otherwise. Given a proper estimator for the discrepancy, this can improve the quality of generated samples by rejecting unrealistic ones. For a given generated sample $x = G(z, c_x)$ with the corresponding class $c_x$, the GOLD rejection sampling is defined as using the following acceptance rate:

$$r(x) := \frac{1}{M} \exp\left(d_{\text{bal}}(x, c_x)\right) = \frac{1}{M} \exp\left(\log \frac{D_G(x)}{1 - D_G(x)} + \frac{\sigma_G}{\sigma_C} \log D_C(c_x|x)\right),$$

(6)

where $M$ is set to be the maximum of $\exp(d_{\text{bal}}(x, c_x))$ among samples. This helps in recovering the true data distribution $p_{\text{data}}(x, c)$, although the model distribution $p_g(x, c)$ is suboptimal.[7]

While the recent work [2] studies a rejection sampling for unconditional GANs, we focus on improving cGANs and our formula (6) of the acceptance rate is different. We also remark that in order to avoid extremely low acceptance rates, following the strategy in [2], we first pullback the ratio with $f^{-1}(r(x))$ ($f$ is the sigmoid function), subtract a constant $\gamma$, and pushforward to $f(f^{-1}(r(x)) - \gamma)$. As in [2], we set the constant $\gamma$ to be a $p$-th percentile of the batch, where $p$ is tuned for datasets. Note that $\gamma$ controls the precision-recall trade-off [45] of samples, as the low acceptance rate (high $\gamma$) improves the quality and the high acceptance rate (low $\gamma$) improves the diversity.

**Active learning.** The goal of active learning [48] is to reduce the cost of labeling by predicting the best real samples (*i.e.*, queries) to label to improve the current model. In training cGANs with active learning, it is natural to find and label samples with high GOLD values since they can be viewed as under-estimated ones with respect to the current model. For unlabeled samples, however, we do not have access to ground-truth class $c_x$ and thus $d(x, c_x)$ (or $d_{\text{bal}}(x, c_x)$). To tackle this issue, we take an expectation of $c_x$ over the class probability using $D_C$ and estimate the conditional term as

$$-\log D_C(c_x|x) \approx \mathbb{E}_{c\sim D_C(c|x)}[-\log D_C(c|x)] = \mathcal{H}[D_C(c|x)],$$

(7)

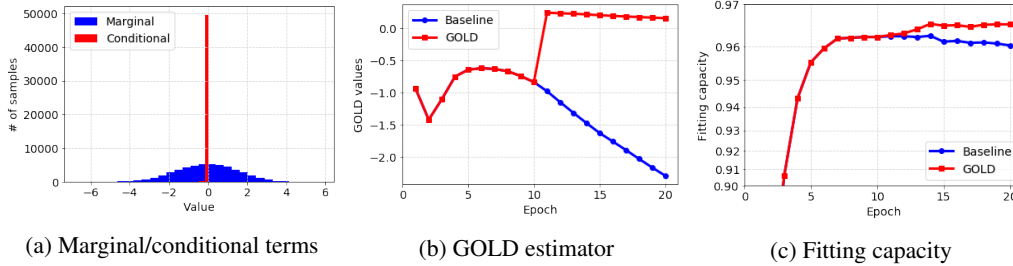

| (a) Marginal/conditional terms | (b) GOLD estimator | (c) Fitting capacity |

Figure 1: (a) Histogram of the marginal/conditional terms of the GOLD estimator. Training curve of the mean of the GOLD estimator (of generated samples) (b) and the fitting capacity (c), for the baseline model and that trained by the re-weighting scheme (GOLD) under MNIST dataset.

where $\mathcal{H}$ is the entropy function. Using the approximation above, the GOLD estimator for the unlabeled real samples can be defined as:

$$d_{\mathtt{unlabel}}(x) := \log \frac{D_G(x)}{1 - D_G(x)} + \mathcal{H}[D_C(c|x)], \tag{8}$$

$$d_{\mathtt{unlabel-bal}}(x) := \log \frac{D_G(x)}{1 - D_G(x)} + \frac{\sigma_G}{\sigma_C} \cdot \mathcal{H}[D_C(c|x)], \tag{9}$$

where $\sigma_G$ and $\sigma_C$ are the standard deviations of marginal and conditional (*i.e.*, entropy) terms.

As in the conventional active learning for classifiers, one can view the first term $\log \frac{D_G(x)}{1-D_G(x)}$ in (8) as a density (or representativeness) score [14, 50], which measures how well the sample $x$ represents the data distribution. The second term $\mathcal{H}[D_C(c|x)]$ is an uncertainty (or informativeness) score [13, 3], which measures how informative the label $c$ is for the current model. Hence, our method can be interpreted as a combination of the density and uncertainty scores [19] in a principled, yet scalable way. We finally remark that we also utilize all unlabeled samples in the pool to train our model, *i.e.*, semi-supervised learning, which can be naturally done in the cGAN framework of our interest.

## 4 Experiments

In this section, we demonstrate the effectiveness of the GOLD estimator for three applications: example re-weighting, rejection sampling, and active learning. We conduct experiments on one synthetic point dataset and six image datasets: MNIST [25], FMNIST [54], SVHN [36], CIFAR-10 [23], STL-10 [11], and LSUN [55]. The synthetic dataset consists of random samples drawn from a Gaussian mixture with 6 clusters, where we assign the clusters binary labels to obtain 2 groups of 3 clusters (see Figure 3). As the choice of cGAN models to evaluate, we use the InfoGAN [9] model for 1-channel images (MNIST and FMNIST), the ACGAN [39] model for 3-channel images (SVHN, CIFAR-10, STL-10, and LSUN), and the GAN model of [16] with an auxiliary classifier for the synthetic dataset. For all experiments, the spectral normalization (SN) [32] is used for more stable training. We set the balancing factor to $\lambda_c = 0.1$ in most of our experiments but lower the value when training cGANs on small datasets.[8] For all experiments on example re-weighting and rejection sampling, we choose the default value $\lambda_c = 0.1$. For experiments on active learning, we choose $\lambda_c = 0.01$ and $\lambda_c = 0$ for synthetic/MNIST and FMNIST/SVHN datasets, respectively. The reported results are averaged over 5 trials for image datasets and 25 trials for the synthetic dataset.

As the evaluation metric for data generation, we choose to use the fitting capacity recently proposed in [41, 27]. It measures the accuracy of the real samples under a classifier trained with generated samples of cGAN, where we use LeNet [25] as the classifier.[9] Intuitively, fitting capacity should match to the 'true' classifier accuracy (trained with real samples) if the model distribution perfectly matches to the real distribution. It is a natural evaluation metric for cGANs, as it directly measures the performance of *conditional generation*. Here, one may also suggest other popular metrics, *e.g.*, Inception score (IS) [46] or Fréchet Inception distance (FID) [17], but the work of [41] have recently shown that when IS/FID of generated samples match to those of real ones, the fitting capacity is

Table 1: Fitting capacity (%) [41] for example re-weighting under various datasets.

|  | MNIST | FMNIST | SVHN | CIFAR-10 | STL-10 | LSUN |
|---|---|---|---|---|---|---|
| Baseline | $96.43_{\pm0.17}$ | $77.97_{\pm1.24}$ | $74.43_{\pm0.71}$ | $36.76_{\pm0.99}$ | $36.73_{\pm0.64}$ | $26.35_{\pm0.82}$ |
| GOLD | $\mathbf{96.62}_{\pm\mathbf{0.15}}$ | $\mathbf{78.34}_{\pm\mathbf{1.11}}$ | $\mathbf{76.71}_{\pm\mathbf{0.94}}$ | $\mathbf{37.06}_{\pm\mathbf{1.38}}$ | $\mathbf{37.65}_{\pm\mathbf{0.71}}$ | $\mathbf{28.21}_{\pm\mathbf{0.86}}$ |

Table 2: Fitting capacity (%) for example re-weighting under various levels of supervision.

|  | Dataset | 1% | 5% | 10% | 20% | 50% | 100% |
|---|---|---|---|---|---|---|---|
| Baseline | SVHN | $72.41_{\pm1.30}$ | $72.99_{\pm1.65}$ | $73.15_{\pm0.96}$ | $73.18_{\pm1.28}$ | $74.04_{\pm1.26}$ | $74.33_{\pm0.71}$ |
| GOLD |  | $\mathbf{75.01}_{\pm\mathbf{1.93}}$ | $\mathbf{75.58}_{\pm\mathbf{0.86}}$ | $\mathbf{75.78}_{\pm\mathbf{0.74}}$ | $\mathbf{76.04}_{\pm\mathbf{1.93}}$ | $\mathbf{76.25}_{\pm\mathbf{1.40}}$ | $\mathbf{76.71}_{\pm\mathbf{0.94}}$ |
| Baseline | CIFAR-10 | $17.99_{\pm0.78}$ | $18.42_{\pm0.71}$ | $21.84_{\pm1.14}$ | $23.13_{\pm1.95}$ | $35.41_{\pm1.03}$ | $36.76_{\pm0.99}$ |
| GOLD |  | $\mathbf{18.28}_{\pm\mathbf{0.65}}$ | $\mathbf{19.15}_{\pm\mathbf{0.97}}$ | $\mathbf{21.91}_{\pm\mathbf{2.56}}$ | $\mathbf{23.89}_{\pm\mathbf{2.02}}$ | $34.95_{\pm1.11}$ | $\mathbf{37.06}_{\pm\mathbf{1.38}}$ |

often much lower than the real classifier accuracy (*i.e.*, low correlation between IS/FID and fitting capacity). Furthermore, IS/FID are not suitable for non-ImageNet-like images, *e.g.*, MNIST or SVHN. Nevertheless, we provide some FID results in Supplementary Material for the interest of readers.

## 4.1 Example re-weighting

We first evaluate the effect of the re-weighting scheme using the loss (5). We train the model for 20 and 200 epochs for 1-channel and 3-channel images, respectively. We use the baseline loss (1) for the first half of epochs and the re-weighting scheme for the next half of epochs. We simply choose $\beta = 1$ for the discriminator loss and $\beta = 0$ for the generator loss. This is because a large $\beta$ for the generator loss unstabilizes training by incurring high variance of gradients.[10] We train the LeNet classifier (for fitting capacity) for 40 epochs, using 10,000 newly generated samples for each epoch. Figure 1b and Figure 1c report the training curves of the GOLD estimator (of generated samples) and the fitting capacity respectively, under MNIST dataset. Figure 1b shows that the GOLD estimator under the re-weighting scheme stably converges to zero, while that of baseline model monotonically decreases. As a result, in Figure 1c, one can observe that the re-weighting scheme improves the fitting capacity, while that of the baseline model become worse as training proceeds. Table 1 and Table 2 report the fitting capacity for fully-supervised settings (*i.e.*, use full labels of datasets to train cGANs) and semi-supervised settings (*i.e.*, use only $x\%$ supervision of datasets to train cGANs), respectively. In most reported cases, our method outperforms the baseline model. For example, ours improves the fitting capacity from 74.43 to 76.71 (+3.06%) under the full labels of SVHN.

## 4.2 Rejection sampling

Next, we demonstrate the effect of the rejection sampling. We use the model trained by the original loss (1) with fully labeled datasets.[11] To emphasize the sampling effect, we use the fixed 50,000 samples instead of re-sampling for each epoch. We use $p = 0.1$ for 1-channel images, and $p = 0.5$ for 3-channel images. Table 3 presents the fitting capacity of the rejection sampling under various datasets. Our method shows a consistent improvement over the baseline (random sampling without rejection), *e.g.*, ours improves from 73.58 to 75.06 (+2.01%) for SVHN. We also study the effect of $p$, the control parameter of the acceptance ratio for the rejection sampling (high $p$ rejects more samples). As high $p$ harms the diversity and low $p$ harms the quality, we see the proper $p$ (*e.g.*, 0.5 for CIFAR-10) shows the best performance. Table 4 and Figure 5 in Supplementary Material present the fitting capacity and the precision and recall on distributions (PRD) [45] plot, respectively, under CIFAR-10 and various $p$ values. Indeed, both low ($p = 0.1$) and high ($p = 0.9$) values harm the performance, and $p = 0.5$ is of the best choice among them.

We also qualitatively analyze the effect of the rejection sampling. The first row of Figure 2 visualizes the generated samples with high marginal, conditional, and combined (GOLD) values. We observe that the random samples (without rejection) often contain low-quality samples with uncertain and/or wrong classes. On the other hand, samples with high marginal values improve the quality (or vividness), and samples with high conditional values improve the class accuracy (but loses the diversity). The samples with high GOLD values get the best of the both worlds, and produce diverse images with only a few wrong classes.

Table 3: Fitting capacity (%) for rejection sampling under various datasets.

|          | MNIST | FMNIST | SVHN | CIFAR-10 | STL-10 | LSUN |
|----------|-------|--------|------|----------|--------|------|
| Baseline | $96.05_{\pm0.41}$ | $77.94_{\pm0.83}$ | $73.58_{\pm0.72}$ | $35.15_{\pm0.51}$ | $34.33_{\pm0.30}$ | $26.43_{\pm0.14}$ |
| GOLD     | $\mathbf{96.17}_{\pm0.63}$ | $\mathbf{78.25}_{\pm0.30}$ | $\mathbf{75.06}_{\pm0.71}$ | $\mathbf{35.98}_{\pm1.15}$ | $\mathbf{35.21}_{\pm1.02}$ | $\mathbf{26.79}_{\pm0.42}$ |

Table 4: Fitting capacity (%) for rejection sampling under CIFAR-10 and various $p$ values.

| Baseline | p = 0.1 | p = 0.3 | p = 0.5 | p = 0.7 | p = 0.9 |
|----------|---------|---------|---------|---------|---------|
| $35.15_{\pm0.51}$ | $35.80_{\pm0.42}$ | $35.87_{\pm0.61}$ | $\mathbf{35.98}_{\pm1.15}$ | $35.85_{\pm0.53}$ | $35.33_{\pm0.53}$ |

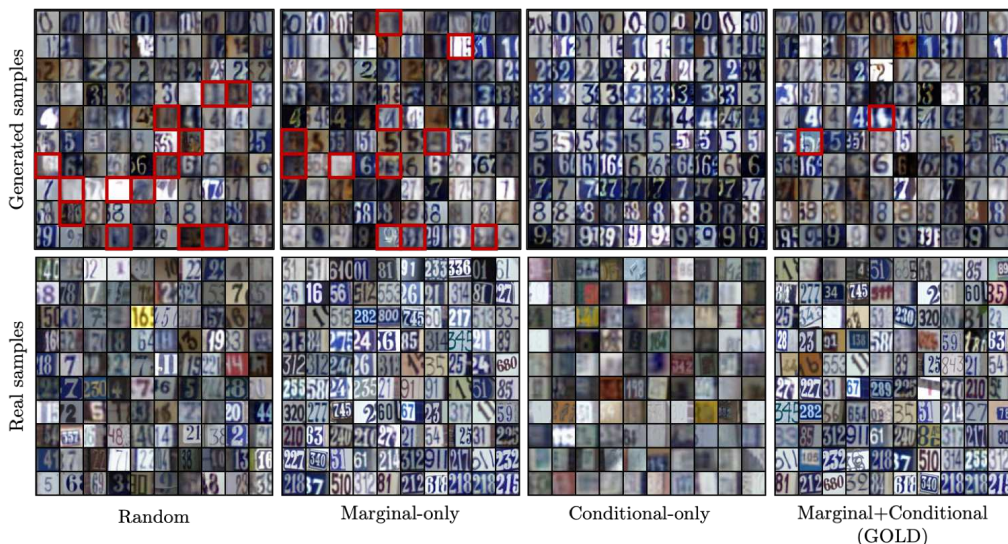

Figure 2: Generated and real samples with high marginal, conditional, and combined (GOLD) values. Generated samples are aligned by the class (each row), and the red box indicates the uncertain and/or wrong classes. See Section 4.2 and Section 4.3 for the detailed explanation.

## 4.3 Active learning

Finally, we demonstrate the active learning results. We conduct our experiments on a synthetic dataset and 3 image datasets (MNIST, FMNIST, SVHN). We train on the semi-supervised setting, as we have a large pool of unlabeled samples. We run 4 query acquisition steps (*i.e.*, 5 training steps), where the triplet of initial (labeled) training set size, query size, and the final (labeled) training set size are set by (4,1,8), (10,2,18), (20,5,40), and (20,20,100) for synthetic, MNIST, FMNIST, and SVHN, respectively. We train the model for 100 epochs, and choose the model with the best fitting capacity on the validation set (of size 100), to compute the GOLD estimator for the query acquisition. Interestingly, we found that keeping the parameters of the generator (while re-initializing the discriminator) for the next model in the active learning scenario improves the performance. This is because the discriminator is easily overfitted and hard to escape from the local optima, but the generator is relatively easy to spread out the generated samples. We use this re-initialization scheme (*i.e.*, keep $G$ and re-initialize $D$) for all active learning experiments. For query acquisition, we use the vanilla version of the GOLD estimator (8) for image datasets, but use the balanced version (9) for the synthetic dataset, as the synthetic dataset suffers from the over-confidence problem.

Figure 3 visualizes the selected queries based on the GOLD estimator under the synthetic dataset. The GOLD estimator has high values on the uncovered or the uncertain (*i.e.*, samples are not obtained) regions, in which high marginal and conditional values occur, respectively. See the leftmost region of column 2 and the upmost region of column 3 for each case. Indeed, both components of the GOLD estimator contribute to the query selection. Consequently, the GOLD estimator effectively selects queries and learn the true joint distribution. In contrast, the random selection often picks redundant or less important regions, which makes the convergence slower. Figure 4 presents the quantitative results. Our method outperforms the random query selection, *e.g.*, the final fitting capacity of our method on MNIST is 94.60, which improves 92.65 of the baseline by 2.10%.

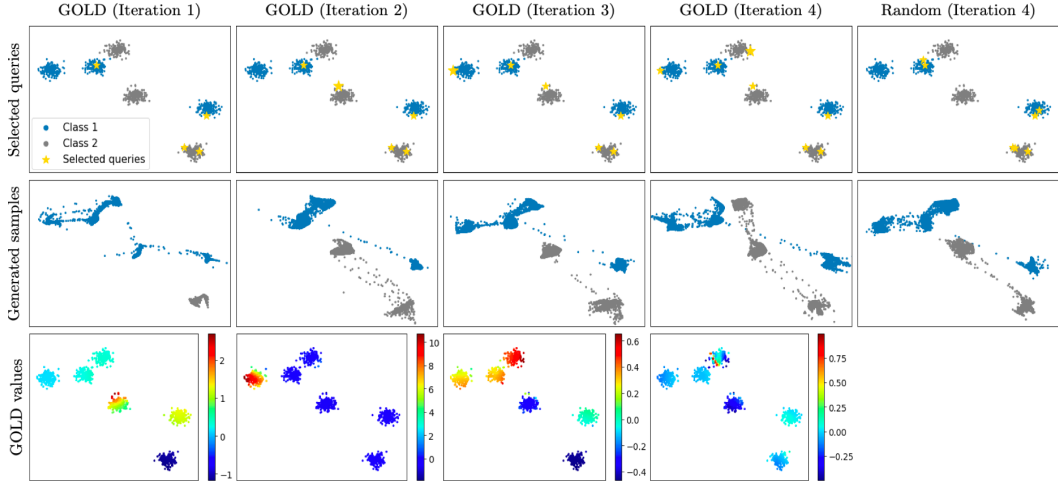

Figure 3: Visualization of the query selection based on the GOLD estimator. The first and second row are selected queries and generated samples, respectively. The third row is the GOLD estimator values, that the sample with the highest value is selected for the next iteration.

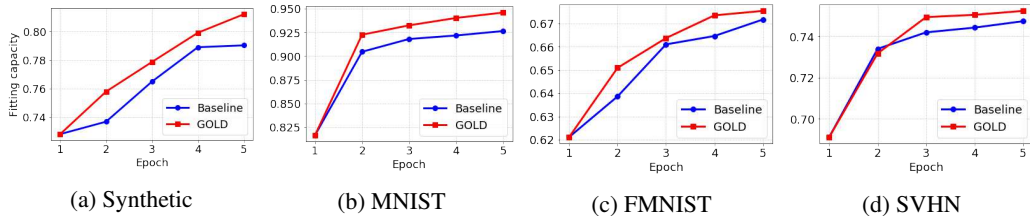

(a) Synthetic　　　　　(b) MNIST　　　　　(c) FMNIST　　　　　(d) SVHN

Figure 4: Fitting capacity for active learning under various datasets.

In addition, we qualitatively analyze the effect of two (marginal and conditional) terms of the GOLD estimator. The second row of Figure 2 presents the real samples with high marginal, conditional, and combined (GOLD) values. We observe that samples picked under high marginal values have multiple digits (which are hard to generate) and those picked under high conditional values have uncertain classes. On the other hand, the GOLD estimator picks the uncertain samples with multiple digits, which takes the advantage of both.

## 5　Discussion and Conclusion

We have proposed a novel, yet simple GOLD estimator which measures the discrepancy of the data distribution and the model distribution on given samples, which can be efficiently computed under the conditional GAN (cGAN) framework. We also propose three applications of the GOLD estimator: example re-weighting, rejection sampling, and active learning, which improves the training, inference, and data selection of cGANs, respectively. We are the first one studying these problems of cGAN in the literature, while those of classification models or the (original unconditional) GAN have been investigated in the literature. First, re-weighting [43] or re-sampling [6, 22] examples are studied to improve the performance, convergence speed, and/or robustness of the convolutional neural networks (CNNs). From the line of the research, we show that the re-weighting scheme can also improve the performance of cGANs. To this end, we use the higher weights for the samples with the larger discrepancy, which resembles the prior work on the hard example mining [49, 28] for classifiers/detectors. Designing a better re-weighting scheme or a better scheduling technique [4, 24] would be an interesting future research direction. Second, active learning [48] has been also well studied for the classification models [13, 47]. Finally, there is a recent work which proposes the rejection sampling [44] for the original (unconditional) GANs [2]. In contrast to the prior work, we focus on the *conditional generation*, *i.e.*, consider both the generation quality and the class accuracy. We finally remark that investigating other applications of the GOLD estimator, *e.g.*, outlier detection [26] or training under noisy labels [43], would also be an interesting future direction.

**Acknowledgments**

This research was supported by the Information Technology Research Center (ITRC) support program (IITP-2019-2016-0-00288), Next-Generation Information Computing Development Program (NRF-2017M3C4A7069369), and Institute of Information & communications Technology Planning & Evaluation (IITP) grant (No.2017-0-01779, A machine learning and statistical inference framework for explainable artificial intelligence), funded by the Ministry of Science and ICT, Korea (MSIT). We also appreciate GPU support from Brain Cloud team at Kakao Brain.

## Footnotes

[2] Projection discriminator [33] is of type (a), but it decomposes the marginal and conditional terms in their architecture. It results another estimator form of the gap of log-densities.

[3] (a) requires some modifications in the architecture and/or the loss function [52, 30].

[4] In experiments, we use the non-saturating GAN loss [15] to improve the stability in training.

[5] We measure the gap of *log*-densities, since it leads to a computationally efficient estimator.

[6] The discriminator $D_G$ is trained to predict 0 and 1 for generated and real samples, respectively.

[7] One may use advanced sampling strategy, *e.g.*, Metropolis-Hastings GAN (MH-GAN) [53]. As MH-GAN requires the density ratio information $p_{\text{data}}/p_g$ to run, one can naturally apply the GOLD estimator.

[8] This is because the generator is more likely to produce bad samples with incorrect attributes for small datasets, which strengthens the wrong signal.

[9] We use training data to train ACGAN and test data to evaluate the fitting capacity, except LSUN that we use validation data for both training and evaluation due to the class imbalance of the training data.

[10] We do not make much effort in choosing $\beta$ as the choice $\beta \in \{0, 1\}$ is enough to show the improvement.

[11] One can also use the model trained by the re-weighting scheme of loss (5) for further improvement.

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
