[Reviews · NeurIPS 2019]

Reviewer 1



This paper revisited the conception of conditional GAN and proposed to make a deeper analysis of the discrepancy between the data distribution and the model distribution. They measured this discrepancy with the gap of log-density of them. Then they derived an approximated surrogate to implement the gap of log-density by quantifying the contributions of faked samples and labelled samples under the framework of ACGAN. This paper make a clear introduction of their work with some originality and quality. However, one of my concerns is the basic assumption, they choose the gap of log-density as a measure of discrepancy but why not others, for example the relative entropy or other generalized metrics.

Reviewer 2



Section 2, line 62 - what is c) , what method groups does it refer to? Sect. 3.2 line 129-131 are written unclear, a better explanation might help for the intuition of the equation (5). "CGAN WITH PROJECTION DISCRIMINATOR", Miyato 2018, are another type of conditional GAN, which is different than the (a) and (b) types from Section 2. They also decompose the loss of a GAN into marginal and conditional ratios, which is part of the GOLD motivator definition. A recent paper "Metropolis-Hastings (MH) Generative Adversarial Networks",Turner2018 uses the discriminator values to create a sampling procedure for better inference in GANs. Since the current paper uses rejection sampling (which can be inferior to MH) it can be good to discuss whether GOLD can work with MH and what performance to expect -- e.g. put this in Figure 2 in the experiments. Discriminator calibration is also a relevant concept from that paper. Sec. 3.2 equation (7) -- please discuss in more detail why the entropy is used to estimate unknown classes, and how this related to uncertainty in the prediction of the most probable class. Experiments 4, line 180 - why are 3 different GAN architectures used, one can do every experiment with ALL 3 of the chosen architectures? Or are there some limitations for data and model? Sec. 4.3 line 245 -- the G and D training scheme re-initialization seems heuristic, the intuition can be better explained line 255 -- it is unclear what is shown in column 2 and 3 in Figure 4. please clarify

Reviewer 3



Clarity - The paper is very well written and very clearly structured - Experimental setup is clear and results are well explained Originality While there are several related methods that use the discriminator for estimating likelihood ratios (e.g., [1], [2], and [3]), the proposed method is specific for the conditional case, and is applied in a new way for modifying training and active learning. The paper clearly states that for rejection sampling is an extension of a similar approach for the unconditional case. In terms of novelty, I think the paper passes the required bar. Quality - The method used is sound. I think the paper does a good job in addressing the main issues that can arise in the proposed method, such as using sufficiently trained discriminators (line 132-133). - That being said, I think the paper should address better some failure cases, such as the effect of GOLD when training is unstable, e.g. divergence or mode collapse. - Experimental results are generally convincing. I appreciate that the paper applied example re-weighting on 6 image datasets and showed improvement of performance in all of them. - While the paper provides clear justification of using the "fitting capacity" for quantitative evaluation, I still believe that providing FID scored would strengthen results. - One main concern I have with using GOLD in re-weighting examples is that it forces the generator to produce samples that can "fool" a classifier to produce the right class rather than generating a more "realistic" example. I think the paper should address this issue in more detail. Significance - The paper proposes a method can potentially improve research in conditional GANs Minor issues: - The GAN loss in Eq. 1 is wrong. The code seems to be using minimax-GAN (i.e., BCE loss for both generator and discriminator). References [1] https://arxiv.org/abs/1606.00709 [2] https://openreview.net/pdf?id=B16Jem9xe [3] https://openreview.net/pdf?id=rkTS8lZAb ==================== Post rebuttal: Thanks for responding thoroughly to my comments. I think this is a good paper, and I vote for accepting it.

[Author Response · NeurIPS 2019]

We thank all the reviewers for their valuable comments, efforts, and time. In particular, we sincerely appreciate that
all reviewers agree on novelty/originality of our method, which we believe a useful contribution to the literature on
conditional GANs. Below are our responses to the reviewers, which we will incorporate in the final manuscript.

──────────────────────────────────── **For Reviewer #1** ────────────────────────────────────

**Justification on the gap of log-densities.** We emphasize that our goal is to measure the point-wise (or sample-wise)
discrepancy, while the relative entropy (that you suggested) measures the discrepancy under the entire domain, *i.e.*, an
expectation of the point-wise discrepancies.[1] To this end, as we did, it is natural to use the difference of the densities for
the point-wise discrepancy. In particular, we suggest to measure the difference of *log*-densities because it leads to a
computationally efficient estimator (as shown in our paper). We will clarify these in the final manuscript.

**Properties of the gap of log-densities.** The gap of log-densities is *not* a metric in a mathematical sense, *i.e.*, it is
neither a function $X \times X \to [0, \infty)$ of a domain $X$, nor satisfies the three conditions you mentioned. Instead, it is
a simple scalar value to measure the point-wise discrepancy. which is enough for our purpose. We will revise our
manuscript to avoid such confusions in using the term 'metric'.

**Why equation (6)?** By using the generator's distribution $p_g$ as a proposal distribution and $p_{\text{data}}/M p_g$ as an acceptance
rate, it is easy to prove that the sample distribution theoretically converges to the target distribution $p_{\text{data}}$. As the GOLD
estimator approximates $\log(p_{\text{data}}/p_g)$, the equation (6) is a natural choice for the acceptance rate.

──────────────────────────────────── **For Reviewer #3** ────────────────────────────────────

**Projection discriminator.** Our definition of type (a) includes the projection discriminator, as we cite the paper in line
61. As R3 mentioned, it decomposes the marginal and conditional terms in their architecture, which would result in
another estimator form of the gap of log-densities. We will add a related discussion to the final draft.

**Metropolis-Hastings GAN (MH-GAN).** As MH-GAN requires the density ratio information $p_{\text{data}}/p_g$ to run, one can
indeed apply the GOLD estimator to it. We will add a related discussion in the final manuscript.

**Other comments.** We will revise our manuscript by clarifying all of the following points.

- Line 62. "$c$)" refers to the codomain of the discriminator $D$, not a method group.
- Line 129-131. The re-weighted loss encourages the generator to strengthen under-generated regions while regularizing
over-generates regions.
- Equation (7). As we do not know the true class $c_x$ of data $x$, we estimate it by $c_x \sim D_C(c|x)$. Taking an expectation
over the class probability leads to the entropy formula (7).
- Line 180. We choose different architectures for different types of inputs since it is more adequate to test a larger
network for a more complex task. To this end, we follow the same choices in the related work, *e.g.*, InfoGAN is used
for the MNIST dataset (1-channel images) whereas ACGAN is for the CIFAR dataset (3-channel images).
- Line 245. The discriminator is often overfitting and thus re-initializing parameters can help to find better local minima.
- Line 255. In column 2, the GOLD value is the highest for the leftmost region, which is uncovered by the generator. In
column 3, the GOLD value is the highest for the upmost region, where samples are not obtained, *i.e.*, uncertain.

──────────────────────────────────── **For Reviewer #4** ────────────────────────────────────

**FID scores and unstable training.** Following R4's suggestion, we will add FID scores and corresponding discussions
in the final manuscript. Here we show some of them. (a) When applied to rejection sampling, our method consistently
improves the FID score as shown in the table below. (b) In order to address R4's question on the case when training is
highly unstable, we also measure FID scores during training with only 10 labeled samples (and no additional unlabeled
samples) from the MNIST dataset. In this case, we observe that mode collapsing occurs at around 10-th epoch, and we
apply our method for the next 10 epochs. The figure on the right shows that our method
can mitigate the instability issue, significantly improving the FID score during training.

|          | MNIST            | FMNIST           | SVHN            | CIFAR-10        | STL-10           | LSUN            |
|----------|------------------|------------------|-----------------|-----------------|------------------|-----------------|
| Baseline | $10.78_{\pm 0.04}$ | $12.38_{\pm 0.03}$ | $8.28_{\pm 0.07}$ | $9.46_{\pm 0.04}$ | $14.47_{\pm 0.04}$ | $14.38_{\pm 0.03}$ |
| GOLD     | $\mathbf{10.70}_{\pm 0.05}$ | $\mathbf{12.32}_{\pm 0.06}$ | $\mathbf{8.12}_{\pm 0.06}$ | $\mathbf{9.44}_{\pm 0.02}$ | $\mathbf{14.44}_{\pm 0.04}$ | $\mathbf{14.35}_{\pm 0.06}$ |

**Realistic generation?** Both our re-weighting scheme and the original one do not force the generator to "fool" a
classifier, *i.e.*, the generator and discriminator are adversarially trained only to force to generate realistic samples and do
not compete to produce the right class. The re-weighting scheme targets both class and reality of samples to improve.

**GAN loss (equation (1)).** We use the non-saturating GAN loss (proposed in the original paper of GAN by Goodfellow
et al.) to improve the stability in training. We will clarify this in the final manuscript to be consistent with our code.

**ImageNet experiments.** Following R4's suggestion, we will add larger-scale experiments to the final manuscript.

taking expectation of the gap of log-densities under the real and generated distributions, respectively.

## Footnotes

[1]For R1's interest, we remark that one can obtain the KL divergence $D_{\text{KL}}(p_{\text{data}}\|p_g)$ and the reverse KL $-D_{\text{KL}}(p_g\|p_{\text{data}})$ by


[Meta-Review · NeurIPS 2019]

After rebuttal, the authors have addressed the comments from the reviewers. All reviewers agree the quality of this paper reaching the standard of NeurIPS.